# Optimizable Control Barrier Functions to Improve Feasibility and Add Behavior Diversity while Ensuring Safety

## Shilei Li *, Zhimin Yuan, Yun Chen, Fang Luo, Zhichao Yang, Qing Ye, Wei Fu and Yu Fu

Department of Information Security, Naval University of Engineering, Wuhan 430033, China
* Correspondence: leeshilei@aliyun.com

**Abstract:** Ensuring safety while retaining maximum performance is a basic requirement for automatic cyber-physical systems, especially for safety-critical applications. A quadratic programming optimization framework called MPC-CBF has recently been presented, which directly unifies model predictive control (MPC) with control barrier functions (CBFs) over the prediction time horizon. However, the conservative nature of CBFs can lead to feasibility problems in real applications. Based on the analysis of the role of the decay rate and the conservative accumulation phenomenon in standard CBF formulations, this paper proposes to directly optimize CBF constraints within the MPC framework. By regarding CBFs as a safety restriction level indicator and an optimizable constraint within the MPC framework, the trade-off between feasibility and safety can be adaptively optimized. The proposed Optimizable CBF (OCBF) model removes the hyper-parameters selection problem in standard CBFs and can adaptively adjust the safety restriction level and increase behavior diversity by adding the corresponding objects in the cost function in MPC. To eliminate the accumulation effects of actual values of the CBF constraints in previous time steps, this paper further proposes a General OCBF (GOCBF) formulation. Compared with existing formulations, the safety margin defined in our GOCBF has intuitive physical meanings and thus provides a more flexible and intuitive mechanism to compromise different objects in terms of ensuring safety while not undermining basic feasibility. Experimental results demonstrate that our algorithm provides a more flexible and intuitive mechanism to achieve this, thus improving feasibility and adding behavior diversity in the MPC-CBF framework.

**Keywords:** optimizable control barrier function; feasibility and safety; safety-critical systems; model predictive control; restriction level indicator

## 1. Introduction

Safety is one of the fundamental problems in many cyber-physical systems. Safety-critical applications, such as auto-driving, motion planning, and mobile robots, have many safety requirements including the range constraints of physical entities (e.g., sensors, actuators) or system variables (e.g., states, inputs, outputs), which require the system to remain in certain sets constantly. These requirements can be described as finding a controller that makes the system achieve certain task-related performances as well as possible, while also respecting various and different safety limits during the whole execution of the task. A common approach is to formulate this as an optimization problem by considering these different or occasionally conflicting constraints simultaneously. Model predictive control (MPC) [1], which repeatedly solves a constrained finite-horizon optimal control problem by taking future state predictions into account, is a popular way to solve this problem. In traditional MPC formulation, safety and any other constraints are usually described as range or inequality limits directly. However, solving a nonlinear optimization in real time remains challenging, because it is difficult to predict whether the system will remain within safety limits when applying the calculated control input. Recent studies have formulated this problem using control barrier functions (CBFs) [2,3], which can avoid

nonlinear optimization by selecting a desired safety set and constructing a corresponding feedback policy that ensures the system remains in the safety set indefinitely. Using CBFs, we can explicitly formulate constraints in the control space to guarantee invariance of a desired safety property. Additionally, the safety (via a control barrier function) and stability (via a control Lyapunov function) can be unified in the context of a quadratic programming (QP) optimization-based controller. The MPC-CBF framework has been widely applied to problems in the area of automotive control [3]. However, task-related performance constraints and CBF constraints may be in conflict in the optimization, and consequently there might not be a solution that satisfies these different constraints simultaneously. What is required is an adaptive tradeoff mechanism to compromise on these different constraints. Furthermore, the possibility of infeasibility increases as more CBF constraints are considered along the prediction time horizon. In particular, because of the conservative nature of standard CBFs, infeasibility issues may come directly from the potential empty intersection between the reachable set and the safe region confined by CBF constraints at each time step in the MPC framework [4,5]. For MPC-CBF, ensuring that the feasible solution space under the CBF constraints is not empty before calculation becomes the main concern. The main difficulty is that there is no simple way of simultaneously enhancing the safety level and the feasible solution space; in other words, the relationship between safety and feasibility may be very complex and cannot be easily determined beforehand. To further increase the applicability of the MPC-CBF framework, it is of great importance that the feasible solution space should be maintained to the best extent.

In this paper, we propose that the safety constraints formulated by CBFs can be regarded as optimizable parameters within the MPC framework rather than as predefined and fixed values. Based on this observation, the paper first proposes an Optimizable CBF (OCBF) and then further proposes a General OCBF (GOCBF) formulation to eliminate the accumulation effects of actual values of the CBF constraints in previous time steps. In this way, the safety restriction level can be adaptively adjusted during the different time steps in the MPC prediction horizon and thus the feasible solution space can be better maintained. Specifically, the main contribution of this article can be summarized as follows.

(1) We propose to directly optimize the CBF constraints within the MPC framework. The proposed model removes the hyper-parameters selection problem in standard CBFs and can adaptively adjust the safety restriction level and increase behavior diversity by adding the corresponding objects in the cost function in MPC. Improved feasibility and behavior diversity can be shown by replacing the predetermined decay rates of CBF constraints with optimizable parameters in various examples.

(2) We propose that the trade-off between safety and feasibility is not only influenced by the decay rates of CBF constraints but also by the accumulation effects of actual values of the CBF in previous time steps. As far as we know, this paper is the first to propose a General and Optimizable CBF formulation (GOCBF) to eliminate the accumulation effects of actual values of the CBF constraints in previous time steps. Compared with existing formulations, the safety margin defined in our GOCBF has intuitive physical meanings and thus provides a more flexible and intuitive mechanism to compromise different objects in terms of ensuring safety without undermining basic feasibility.

(3) We demonstrate the efficacy of the proposed OCBF and GOCBF formulations in various examples.

Following the introduction presented in Section 1, the remainder of this paper is organized as follows: Section 2 presents related works and especially compares research works sharing some similarities with ours; Section 3 provide a brief but necessary background to the proposed technique; Section 4 describes the framework in detail; Section 5 presents and discusses the simulation results of the proposed method; and Section 6 presents our conclusions.

## 2. Related Works

Barrier functions (BFs) are Lyapunov-like functions [6,7], whose use can be traced back to optimization problems [8]. More recently, they have been employed to realize set invariance [9–11] and multi-objective control [12]. First proposed by [2], CBFs are natural extensions of BFs to systems with control inputs and are used to ensure the forward invariance of the safe set. Consequently, they can transform a constraint defined over system states into a constraint on the control input. Through the development of CBFs [13], safe feedback policies can be efficiently computed by solving a quadratic program (QP), and CBFs are becoming increasingly popular in safety-critical applications [3] due to their simplicity of implementation. By predefining a safe set, CBFs are frequently used to verify safety properties, typically working as a supervisory controller on top of a legacy controller in a minimal control intervention manner without involving the difficult task of computing the system's reachable set. CBFs are more computationally affordable than reachability analysis methods [14–16], which handle safety constraints by finding the set of initial states from which there exists a control input that can keep the system within the safe feasible set despite disturbances. CBFs can also be combined with Control Lyapunov Functions (CLFs) [17–19] to further ensure system stability. Besides the continuous-time domain, [20] also generalized the formulation of CBFs into discrete-time systems (DCBF).

For various constraint satisfaction problems, MPC is a prevalent strategy to achieve optimal performance while avoiding greedy behaviors by taking future states prediction into consideration. In standard MPC, the constraint satisfaction problem is formulated as an optimization framework that is repeatedly solved at each time step. The optimization finds an optimal input sequence over a given prediction horizon based on the system dynamic model. Only the first input of that sequence is applied to the real system and the resulting state is used as the starting point for the next optimization process. Many studies use MPC while also considering system safety [21–30]. In the context of MPC, safety requirements are usually directly formulated as constraints such as range constraints or actuation limits. In particular, in the majority of the literature focusing on collision avoidance [24–29], safety distance constraints are directly defined under Euclidean norms, which do not confine the agent's movement unless the agent is relatively close to the obstacle. A larger prediction horizon is often required to make the agent take early active action to avoid obstacles even far away from it, which increases the computational burden in the MPC optimization. To solve this problem, some existing work has attempted to combine the advantages of MPC and CBFs. In the MPC-CBF framework, the safety constraints are represented by the CBFs and thus the prediction horizon can be reduced by the conservativism induced by the CBFs. In [21], the authors use continuous-time CBFs as constraints inside a discrete-time MPC. In [30] MPC is further unified with discrete-time CBFs into one optimization problem. To further reduce the computation burden, Ref. [31] proposes the use of MPC-Generalized CBF (MPC-GCBF), which changes the multi-step horizon optimization problem of CBFs into a single-step optimization formulation; however, a one-step constraint may not sufficiently confine the system. CBF applications have also been extended to systems with a high relative degree [32,33], while additional developments have considered the challenges brought by unknown system disturbances, system model errors, or parameter uncertainties [34–36]. Machine learning techniques have also been applied to achieve adaptive safety for systems with parameter uncertainties [37–41] in a data-driven fashion.

However, despite its popularity, there is no generic method that can guarantee the feasibility of the MPC-CBF model while it is solved online. In particular, CBF often leads to conservative behaviors because the predefined invariance condition limiting the system to remain is only a subset of the whole safety set. The associated QPs can easily be infeasible when both state constraints and tight control bounds are involved. In practice, a feasible solution requires heuristics and parameter tuning.

To address the problem of feasibility and conservativeness issues, this paper first proposes an Optimizable CBF (OCBF) inspired by previous work on MPC-CBFs. The closest to our method is an optimal-decay form of CBF-QP or CLF-CBF-QP [4,5], which

introduced a new variable and related quadratic cost with a tunable hyper-parameter to indirectly optimize the decay rate of the CBF constraints (Optimal-Decay CBF, ODCBF) in the MPC optimization framework. In our method, we directly optimize the decay rates of the CBF constraints without adding any new variables. We further propose a more general form of OCBF (GOCBF), which is not only simpler and more intuitive but can also eliminate the accumulation effects of the actual values of CBF constraints in previous time steps, thus further improving feasibility. Moreover, the proposed method can generate diverse behaviors with different safety restriction levels by adding the corresponding objects with physical meanings into the cost function.

## 3. Problem Formulation and Preliminaries

Considering a nonlinear affine system with the form:

$$\dot{x}(t) = f(x(t)) + g(x(t))u \tag{1}$$

where $x \in \mathbb{R}^n, u \in \mathbb{R}^m$ are, respectively, the state and control input of the system. Meanwhile, $f : \mathbb{R}^n \to \mathbb{R}^n$ is the drift dynamics and $g : \mathbb{R}^n \to \mathbb{R}^{n \times m}$ is the input dynamics. To take into account performance optimality, an infinite time horizon cost function is considered and minimized along with the trajectories of the system (1) which were restricted by various constraints. Commonly, the above safe optimal control problem can be formulated as:

$$\begin{aligned} \min_{u} J(u,x) &= \int_t^\infty c(x(\tau), u(\tau))d\tau \\ s.t. \quad &(1), x(0) = x_0, x \in X, u \in U_{adm}\ (x(t)) \end{aligned} \tag{2}$$

where $x \in X$ and $u \in U_{adm}(x(t))$ represent the acceptable safe set of the system's states and control inputs. The safe set is usually directly represented as range constraints from simple range limits to complex functions, which are derived from actual applications or specific task requirements, such as a robotic arm's actuator limit or the unsafe exploration regions of a mobile robot. The goal here is to design a safe optimal control policy for the system (1) and the cost function $c(x, u)$ in (2) is often defined as:

$$c(x, u) = Q(x) + u^T R u \tag{3}$$

where $Q(x)$ is a positive-definite function and $R$ is a symmetric positive-definite matrix $R = R^T > 0$. Here, $Q(x)$ and $u^T R u$ represent the state-related cost and the control cost, respectively; these are the performance metrics that various methods strive to optimize to the best level possible.

### 3.1. Control Barrier Function

The CBF definition is closely tied to the concept of the control invariant set. A set $S$ is a control invariant set if a controller $\pi : \mathbb{R}^n \to U$ exists such that for all initial conditions $x(0) \in S, \forall t \geq 0, x(t) \in S$. CBFs have recently been introduced as a promising way to ensure set invariance by considering the system dynamics and implying forward invariance of the safe set. Given a closed safe set $C \in \mathbb{R}^n$, it is assumed that the set $C$ is defined as:

$$\begin{aligned} C &= \{x \in \mathbb{R}^n : h(x) \geq 0\} \\ \partial C &= \{x \in \mathbb{R}^n : h(x) = 0\} \\ Int(C) &= \{x \in \mathbb{R}^n : h(x) > 0\} \end{aligned} \tag{4}$$

where $Int(C)$ and $\partial C$, respectively, denote the interior and boundary of the set $C$. Moreover, $h : \mathbb{R}^n \to \mathbb{R}$ is a continuously differentiable function. The function $h$ becomes a CBF if $\frac{\partial h}{\partial x} \neq 0$ for all $x \in \partial C$, and $\forall x \in \{x | h(x) \geq 0\}$, $h$ satisfies [3,13]

$$\exists u \in U \quad s.t. \quad \dot{h}(x, u) \geq -\gamma(h(x)) \tag{5}$$

where $\gamma(\cdot)$ is a class-K function, i.e., $\gamma(\cdot)$ is strictly increasing and satisfies $\gamma(0) = 0$. It is straightforward to see that for a valid CBF, $\{x|h(x) \geq 0\}$ is actually a control invariant set. In other words, if the initial state is within a safe set, the existence of a CBF on that set guarantees its forward invariance. Consequently, using Equation (5), a set of control inputs that keeps the system trajectory safe can be quantified.

The idea of CBF also derives from Control Lyapunov Functions (CLF). Suppose the control objective is to (asymptotically) stabilize the nonlinear control system (1) to a point $x^* = 0$, i.e., driving $x(t) \to 0$. In a nonlinear context, this can be achieved by equivalently finding a feedback control law that drives a positive definite function, $V(x) : \mathbb{R}^n \to \mathbb{R}_{\geq 0}$, to zero. If $\exists u \in U \quad s.t. \quad \dot{V}(x, u) \leq -\alpha(V(x))$, then it can prove that the system is stabilizable to $V(x^*) = 0$, i.e., $x^* = 0$. The design of function $V$ can stabilize the system without the need to explicitly construct the feedback controller; that is, we only need a controller that results in the desired inequality on $\dot{V}$ to exist. Like CLF, it is only necessary to have a controller satisfying Equation (5) to ensure that the system state is always within the safe set. However, it must be pointed out that the safe set restricted by Equation (5) is actually only a subset of the whole safe set defined by $h(x) \geq 0$, which inevitably leads to conservative behaviors in CBFs.

In actual applications, the CBF controller is often used as a supervisory controller that enforces the state to stay inside $\{x|h(x) \geq 0\}$ with the following quadratic programming:

$$u^* = \underset{u \in U}{\operatorname{argmin}} \|u - u^0\|^2$$
$$s.t. \quad \dot{h}(x, u) \geq -\gamma(h(x)) \tag{6}$$

where $u^0$ is the input of the original controller. Equation (6) guarantees the feasibility of the CBF quadratic programming (QP) via an implicitly defined control invariant set. It is particularly formulated in a minimally invasive fashion, i.e., modifying an existing controller in a minimal way to guarantee safety. Therefore, Equation (6) provides a foundational framework for safety critical control using CBFs.

### 3.2. Model Predictive Control with CBF Constraints

MPC is widely used to achieve optimal performance while satisfying different constraints. In standard MPC, a multi-step optimization is calculated to find an optimal control input sequence over a given prediction horizon, based on the system's dynamic model. Corresponding to Equations (2) and (3), the MPC with a prediction horizon of $N$ time steps can be formulated as:

$$\min_u J(u, x) = \min_{u_{t:t+N-1|t}} \left[ \sum \left( Q(x_t) + u_t^T R u_t \right) \right] \tag{7a}$$

$$x_{t+k+1|t} = f(x_{t+k|t}, u_{t+k|t}), k = 0 \cdots N - 1 \tag{7b}$$

$$x_{t+k|t} \in X, u_{t+k|t} \in U_{adm}(x), k = 0 \cdots N - 1 \tag{7c}$$

$$x_{t|t} = x_t, x_{t+N|t} \in X_f \tag{7d}$$

In Equation (7b) describes the system dynamics, (7c) shows the state/input constraints along the horizon, and (7d) represents the initial condition and the expected terminal set constraint. The optimal solution to (7a) at time $t$ is a sequence of control inputs $u_{t:t+N-1|t}^* = [u_{t|t}^*, \cdots, u_{t+N-1|t}^*]$. Nevertheless, only the first control input of that sequence $u_{t|t}^*$ is applied to the system. In other words, in MPC, the feedback control law is given by:

$$u(t) = u_{t|t}^*(x_t) \tag{8}$$

This multi-step optimal control problem (7a–7d) is repeated during MPC, and the resulting state $x_{t+1}$ is used as the starting point for the next optimization cycle. The whole process yields the MPC strategy.

Traditionally, the safety constraints in (7c) are directly described as range constraints, which are called MPC-DC in reference [30] (Direct Distance Constraints expression, rather than CBFs). For safety-critical control systems using MPC-DC, the main difficulty arises from the need to predict whether the system will remain in the safe set when applying the calculated control input. Combined with CBFs, safe feedback policies can be efficiently computed by selecting a desired safety set first and then restricting the corresponding feedback policy to satisfy the system forward invariance indefinitely. Consequently, the MPC-CBF, as a combination of MPC and CBF, can be formulated by replacing the direct constraints expression with the CBF constraints in the MPC Equation (7a–7d). To ensure the stability of the system, the CLF can also be added.

## 4. Adaptive and Flexible Control Barrier Functions through Parameters Optimization

In practice [3–5], $\alpha(\cdot)$ and $\gamma(\cdot)$ in CLF and CBFs are commonly chosen as linear functions with constant coefficient $\alpha$ and $\gamma$, i.e., $\alpha(V(x)) = \alpha V(x)$, $\gamma(h(x)) = \gamma h(x)$. In numerical simulations, $\dot{h}(x, u) = \frac{h_{t+\Delta t} - h_t}{\Delta t}$, thus according to Equation (5), we have:

$$\dot{h}(x, u) = \frac{\Delta h}{\Delta t} = \frac{h_{t+\Delta t} - h_t}{\Delta t} \geq -\gamma h_t \tag{9}$$

$$h_{t+\Delta t} \geq (1 - \gamma \cdot \Delta t) h_t \tag{10}$$

Here, $h_{t+\Delta t}$ and $h_t$ represent the next and current time step value of $h(x)$, respectively, and $\Delta t$ is the simulation step. In particular, Equation (10) can be extended to the discrete-time domain as follows [20]:

$$\Delta h(x_{t+k|t}, u_{t+k|t}) \geq -\gamma h(x_{t+k|t}), k = 0 \cdots N - 1 \tag{11}$$
$$0 < \gamma \leq 1$$

where $\Delta h(x_{t+k|t}, u_{t+k|t}) := h(x_{t+k+1}) - h(x_{t+k})$. Equation (11) can be seen as an implicit form of Equation (9) by eliminating the time step $\Delta t$, which uses a new variable $\gamma$ to replace $\gamma \cdot \Delta t$ as $\Delta t$ is a pre-determined constant. As a result, satisfying Equation (11), we have $h(x_{t+k+1}) \geq (1 - \gamma)h(x_{t+k})$, i.e., the lower bound of $h(x_{t+k+1})$ should never decrease below the product of the rate $1 - \gamma$ and $h(x_{t+k})$ in the discrete-time form.

For MPC-CBF, the main concern is how to guarantee the feasibility of the optimization calculation, i.e., the feasible set under the CBF constraints should not be empty. To clarify the feasibility question, the set of state space satisfying the CBF constraints (11) in MPC framework can be defined as:

$$S_{CBF,k} = \{x \in X : h(x_{t+k+1}) \geq (1 - \gamma)h(x_{t+k})\}, k = 0 \cdots N - 1 \tag{12}$$

**Remark 1.** *In addition to the value of optimal value, which depends on the states and the inputs of previous time steps, $S_{CBF,k}$ is heavily influenced by the value of $\gamma$.*

According to Equation (12), when $\gamma$ is small, such as $\gamma \to 0$, the corresponding $1 - \gamma$ becomes bigger, the restriction on $h(x_{t+k+1})$ becomes stricter, and thus the safe region becomes smaller. Conversely, if $\gamma$ is large, such as $\gamma \to 1$, the corresponding $1 - \gamma$ becomes smaller, the restriction on $h(x_{t+k+1})$ becomes looser, and thus the safe region becomes bigger. Under the same conditions, the bigger the $\gamma$ value is, the looser the restriction on $h(x_{t+k+1})$ will be, and vice versa. However, it should be noted that the relationship between feasibility and the CBF safety restriction level may vary according to different applications. In some situations, a high restriction level may increase the final feasible space, and in others, a low restriction level may increase the feasible space. For example, in collision-avoidance applications, a high restriction level may lead to early active avoiding behaviors, improving feasibility; in contrast, for other applications such as environment exploration or tracking behavior, a low restriction level can reduce the conflict among different objects

and thus improve feasibility. It is difficult or even impossible to predetermine the optimal value of $\gamma$.

**Remark 2.** *$\gamma$ can be used as a restriction level indicator and should be regarded as an optimizable parameter rather than a predetermined hyper-parameter in CBF. If the value of $\gamma$ can be changed adaptively, the restriction level can also be adjusted adaptively to realize a better trade-off between safety and feasibility.*

To further change the decay coefficient, the Optimal-Decay form of CBF (ODCBF) is proposed in reference [4] to change the CBF constraint as follows:

$$h(x_{t+k+1}) \geq \omega_{t+k}(1 - \gamma)h(x_{t+k}), \omega_{t+k} \geq 0 \tag{13}$$

where $\omega_{t+k}$ is the new added variable, which is used as an optimizable variable during each time step to change the decay coefficient of CBF during MPC optimization.

According to Equation (13), if $\omega_{t+k}$ changes, the corresponding product value of $\omega_{t+k}(1 - \gamma)$ will also change, even when $\gamma$ remains as the predetermined constant. Here, in reference [4], $\omega_{t+k}$ is only required to be bigger than zero. Nevertheless, if $\omega_{t+k}$ is too big, it may make the value of $\omega_{t+k}(1 - \gamma) > 1$. If $\omega_{t+k}(1 - \gamma) > 1$, the CBF function will be too conservative and may therefore heavily reduce the feasible space. Furthermore, in reference [4], $\gamma$ is still used as a predetermined hyper-parameter which cannot directly reflect the actual restriction level of CBF. Similarly, for continuous applications, the authors instead use the formulation $\dot{h}(x, u) \geq -\omega \gamma h(x)$ in reference [5], which has the same effect as the discrete Equation (13).

Based on the conception that $\gamma$ should be used as a restriction level indicator which can vary adaptively rather than keep constant during the whole time horizon, we propose to directly set the predetermined $\gamma$ as an optimizable variable $\gamma_t$ in MPC-CBF during each time step, which is optimized to realize the automatic trade-off between safety and feasibility. The proposed Optimizable CBF (OCBF) can be formulated as:

$$\min_{u,\gamma} J(u, x) = \min_{u_{t:t+N-1|t}, \gamma_{t:t+N-1|t}} \left[ \sum (Q(x_t) + u_t^T R u_t + \Phi(\gamma_t)) \right] \tag{14a}$$

$$x_{t+k+1|t} = f(x_{t+k|t}, u_{t+k|t}), k = 0 \cdots N - 1 \tag{14b}$$

$$x_{t+k|t} \in X, u_{t+k|t} \in U_{adm}(x), k = 0 \cdots N - 1 \tag{14c}$$

$$x_{t|t} = x_t, x_{t+N|t} \in X_f \tag{14d}$$

$$h(x_{t+k+1}) \geq (1 - \gamma_{t+k})h(x_{t+k}), k = 0 \cdots N - 1 \tag{14e}$$

Compared with standard MPC-CBF, the $\gamma_{t:t+N-1|t}$ is now regarded as optimizable variables rather than the predefined constant in (14a). In this way, the MPC-CBF optimization framework can increase or decrease the value of $\gamma_t$ as required during different time steps to enhance feasibility.

**Remark 3.** *The desired preference in the trade-off between the safety restriction level and feasibility can be realized flexibly by using different cost formulations of $\Phi(\gamma_t)$ in (14a).*

In Equation (14a), $\Phi(\gamma_t)$ is used as the cost related to $\gamma_t$ which can be optimized to represent different preferences. Three formulations are proposed here:

(1)    Prefer more conservative behaviors

If more emphasis is placed on safety, the value of $\gamma_t$ should be as small as possible. Equation (14a) can be specified as:

$$\min_{u,\gamma} J(u,x) = \min_{u_{t:t+N-1|t},\gamma_{t:t+N-1|t}} \left[ \sum \left( Q(x_t) + u_t^T R u_t + \gamma_t^T W_\gamma \gamma_t \right) \right]$$
$$0 < \gamma_t \leq 1, \tag{15a}$$

Here, the optimizable cost $\Phi(\gamma_t) = \gamma_t^T W_\gamma \gamma_t$. like Matrix $R$, the Matrix $W_\gamma$ is a symmetric positive-definite matrix that determines the relative weights in the entire cost. Through Equation (15a), we can determine the optimized $\gamma_t$ which tends to be as small as possible, thus safety is preferred under the feasibility condition and the final behavior will be more conservative.

(2) Prefer more aggressive behaviors

Conversely, if more aggressive behaviors are preferred, the value of $\gamma_t$ should be as big as possible. So, Equation (14a) can be specified as:

$$\min_{u,\gamma} J(u,x) = \min_{u_{t:t+N-1|t},\gamma_{t:t+N-1|t}} \left[ \sum \left( Q(x_t) + u_t^T R u_t - \gamma_t^T W_\gamma \gamma_t \right) \right]$$
$$0 < \gamma_t \leq 1 \tag{15b}$$

According to (15b), the optimized $\gamma_t$ tends to be as large as possible, thus the optimization will be solved under the minimum safety restriction level requirement and the final behavior will be more aggressive.

(3) Prefer user-defined behaviors

At the same time, the preferred value of $\gamma_t = \gamma_0$ can also be defined. To make the value of $\gamma_t$ incline to $\gamma_0$, Equation (14a) can be specified as:

$$\min_{u,\gamma} J(u,x) = \min_{u_{t:t+N-1|t},\gamma_{t:t+N-1|t}} \left[ \sum \left( Q(x_t) + u_t^T R u_t + (\gamma_t^T - \gamma_0^T) W_\gamma (\gamma_t - \gamma_0) \right) \right]$$
$$0 < \gamma_k, \gamma_0 \leq 1 \tag{15c}$$

According to (15c), during optimization, the optimized $\gamma_t$ tends to be $\gamma_0$, so the trade-off between feasibility and safety is determined by the user-defined value $\gamma_0$.

It should be noted that, although different equation forms can be used for $\Phi(\gamma_t)$, safety is always ensured by the CBF constraints only with different restriction levels. By adaptively optimizing the value of $\gamma_t$ during each time step, the feasibility can be enhanced by changing the safety restriction to a proper level with the preference defined by $\Phi(\gamma_t)$.

**Remark 4.** *The lower bound of the CBF constraint at the next time step $h(x_{t+1})$ is a range determined not only by the decay coefficient $(1 - \gamma_t)$ but also by the CBF value at the current time $h(x_t)$.*

From the discrete time Formulation (11) or (12), if $\gamma_t$ has a constant value at different time steps, we can have the following CBF constraints: $h(x_{t+1}) \geq (1 - \gamma)h(x_t)$, $h(x_{t+2}) \geq (1 - \gamma)h(x_{t+1}), \ldots, h(x_{t+k+1}) \geq (1 - \gamma)h(x_{t+k})$ and consequently, we can have:

$$h(x_{t+k+1}) \geq (1 - \gamma)h(x_{t+k}) \geq (1 - \gamma)^2 h(x_{t+k-1}) \geq \cdots \geq (1 - \gamma)^{k+1}h(x_t) \tag{16}$$

According to Equation (16), it can be clearly seen that the lower bound of the CBF at the next time step is not only influenced by the $\gamma$, but also by the values of the CBF at the previous time steps. It must be pointed out that the current value $h(x_t)$ plays a conservative role in the CBF constraint of the next time step, so the bigger the value of $h(x_t)$, the higher the lower bound of the $h(x_{t+1})$. This phenomenon is contrary to the intuition that $h(x_{t+1})$ would have a bigger variation space if the value of $h(x_t)$ were larger and far from the boundary. Furthermore, because the prediction horizon of the MPC is

limited, $(1 - \gamma)^{k+1}$ cannot reduce to zero at the end of the time horizon. Consequently, a mechanism is required to reduce the accumulated conservativism caused by $h(x_t)$.

To improve the feasibility in MPC-CBF, reference [31] proposes the MPC-Generalized CBF (MPC-GCBF) to change the CBF from a multi-step horizon optimization problem to a one-step optimization formulation:

$$h(x_{t+m|t}) \geq (1 - \gamma)^m h(x_{t|t}) \tag{17}$$

where multi-step constraints in (11) are only posed on two nonadjacent steps in MPC-GCBF. Equation (17) can concurrently enhance feasibility and decrease computational time by reducing multi-step constraints into one-step constraints. However, one-step constraints might not confine the system sufficiently and the optimization problem may still become infeasible after a while in the closed trajectory, as illustrated and discussed in reference [4]. Besides reducing computation time, Equation (17) can also be regarded as changing the decay coefficient from $(1 - \gamma)$ to $(1 - \gamma)^m$. Different values of $m$ produce different values of actual decay coefficient, and consequently, the safety restriction level is adjusted in the end.

Intuitively, when the value of $h(x_t)$ is decreased, the lower bower of CBF can be depressed in the following time steps in MPC. Based on this observation, we propose to further adapt the value of $h(x_t)$ to enhance feasibility despite its real value. To handle the $\gamma$ and $h(x_t)$ variables together, we propose to replace the product $(1 - \gamma_t)h(x_t)$ with a new optimizable variable $S_t$:

$$\begin{aligned} h(x_{t+k|t}) \geq S_{t+k}, k = 1 \cdots N - 1 \\ S_{t+k} \geq 0, k = 1 \cdots N - 1 \end{aligned} \tag{18}$$

Here, the CBF constraint Equation (14e) is revised with a new, more general, and intuitive one. Compared with Equation (14e), Equation (18) not only avoids the conservative accumulation introduced by $h(x_t)$ but also flexibly adapts the safety restriction level according to actual situations to enhance feasibility.

Adding our GOCBF (Equation (18)) into the MPC results in our new MPC framework with General and Optimizable CBF (MPC-GOCBF):

$$\min_{u,S_t} J(u,x) = \min_{u_{t:t+N-1|t}, S_{t:t+N-1|t}} \left[ \sum (Q(x_t) + u_t^T R u_t + \Phi(S_t)) \right] \tag{19a}$$

$$x_{t+k+1|t} = f(x_{t+k|t}, u_{t+k|t}), k = 0 \cdots N - 1 \tag{19b}$$

$$x_{t+k|t} \in X, u_{t+k|t} \in U_{adm}(x), k = 0 \cdots N - 1 \tag{19c}$$

$$x_{t|t} = x_t, x_{t+N|t} \in X_f \tag{19d}$$

$$h(x_{t+k+1}) \geq S_{t+k}, k = 0 \cdots N - 1 \tag{19e}$$

As Equation (14a), the cost part $\Phi(S_t)$ must also be added into the MPC-GOCBF. As with Equation (15c), $\Phi(S_t) = (S^T - S_0)W_S(S - S_0)$ can be set to prefer certain safety restriction level behaviors. Here, $S_0$ can be used as a safety margin with clear physical meanings predetermined by the user. If $S_0 = 0$ is set, our MPC-GOCBF becomes MPC-DC, as the safety constraints are set as $h(x_t) \geq 0$. Consequently, our MPC-GOCBF can be regarded as a generalized formulation of both MPC-CBF and MPC-DC.

In summary, our MPC-OCBF or MPC-GOCBF can be described as Algorithm 1:

---

**Algorithm 1:** The MPC-OCBF or MPC-GOCBF Algorithm

---

**Define** the cost function using Equation (15a) or (15b) or (15c) or (19a)
The task or dynamic-model related constraints using Equation (19b)–(19d)The safety constraints
described by OCBF or GOCBF using Equations (14e) or (19e)
**Initialization:** the corresponding optimization parameters such as $Q, R, W_\gamma$ or the user-preferred
safety level indicator $\gamma_0$ or $S_0$ in the cost function, the initial state of the agent, the MPC horizon
$N$, the time step $\Delta t$
**While** task is not finished
**Optimization** the cost function in the whole time horizon based on the MPC algorithm to
produce the control input sequence $u^*_{t:t+N-1|t}$ and $\gamma_{t:t+N-1|t}$, or $S_{t:t+N-1|t}$
**Update** the agent state using the first one in the control input sequence based on Equation (8)
**End While**

---

During one optimization time step, the time complexity of traditional MPC-CBF is $O(m \times N)$, where $m$ is the dimension of the control input and $N$ is the MPC time horizon. Compared with MPC-CBF, the time complexity of our MPC-OCBF or MPC-GOCBF is increased as $O((m+1) \times N)$ because a new optimizable variable $\gamma$ or $S$ is added. Nevertheless, it should be noted that our MPC-OCBF or MPC-GOCBF can have a smaller $N$ because of its ability to enhance feasibility by compromising different conflicting objects adaptively. For traditional MPC-CBF, its prediction time horizon $N$ cannot be too small, because of feasibility problems as discussed in Section Introduction and Related Works. In this way, we can reduce the time-complexity of our MPC-OCBF or MPC-GOCBF to a same level as MPC-CBF in real applications.

## 5. Experimental Results and Discussions

To evaluate the performance of our OCBF and GOCBF methods, we compare them with the corresponding algorithms in tasks used in mostly related references [4,5,30]. All simulations were run in MATLAB and the optimal control was formulated with Yalmip [42] as the modelling language and solved with IPOPT [43].

### 5.1. Two-Dimensional Double Integrator for Static Obstacle Avoidance

Firstly, we use the static obstacle avoidance problem as an illustrative example. A robot is required to move from a start point to a fixed goal while avoiding a static obstacle. The motion of the robot obeys the linear discrete-time 2D double-integrator formulation defined in reference [30].

$$x_{k+1} = Ax_k + Bu_k \tag{20}$$

The robot is subject to state constraint $X$ and input control constraints $U$:

$$
\begin{aligned}
X &= \left\{ x_k \in \mathrm{R}^4 : x_{\min} \le x_k \le x_{\max} \right\} \\
U &= \left\{ u_k \in \mathrm{R}^2 : u_{\min} \le u_k \le u_{\max} \right\}
\end{aligned} \tag{21}
$$

The MPC-CBF is adopted to realize the obstacle avoidance behavior, and here a quadratic barrier function is used as:

$$h_k = (x_k(1) - x_{obs})^2 + (x_k(2) - y_{obs})^2 - r_{obs}^2 \tag{22}$$

where $x_{obs}, y_{obs}, r_{obs}$ describe the x/y-coordinate and radius of the static obstacle. The cost function in the optimization is formulated as:

$$c(x, u) = x_k' Q x_k + u_k' R u_k + x_N' P x_N \tag{23}$$

Here, we adapt the same parameters as the reference [30]. In Equation (23), $Q = 10 \cdot I_4$, $R = I_2$, and $P = 100 \cdot I_4$. The start and target positions are $(-5; -5)$ and $(0; 0)$, with the location of the static round obstacle $(x_{obs}, y_{obs}) = (-2, -2.25)$ and $r_{obs} = 1.5$. The lower

and upper bounds are $x_{\max}, x_{\min} = \pm 5 \cdot I_{4\times 1}, u_{\max}, u_{\min} = I_{2\times 1}$. The MPC horizon is $N = 8$. The initial state of the robot is set as $X = [-5, -5, 0, 0]$. The results are illustrated in Figures 1–6, where the obstacle is represented by a red circle, and the start and target positions are labelled as the blue and red diamond, respectively.

(1)    Prefer more conservative behaviors

Equation (15a) is used in our MPC-OCBF and the results are compared with MPC-CBF with fixed $\gamma$. For MPC-CBF, two experiments were run with $\gamma = 0.01$ and $\gamma = 0.1$, respectively. For MPC-OCBF, four experiments were run with the weight matrix $W_\gamma = 10^2 \cdot I_N$, $10^3 \cdot I_N$, $10^4 \cdot I_N$ and $10^5 \cdot I_N$, respectively. The results are illustrated in Figure 1.

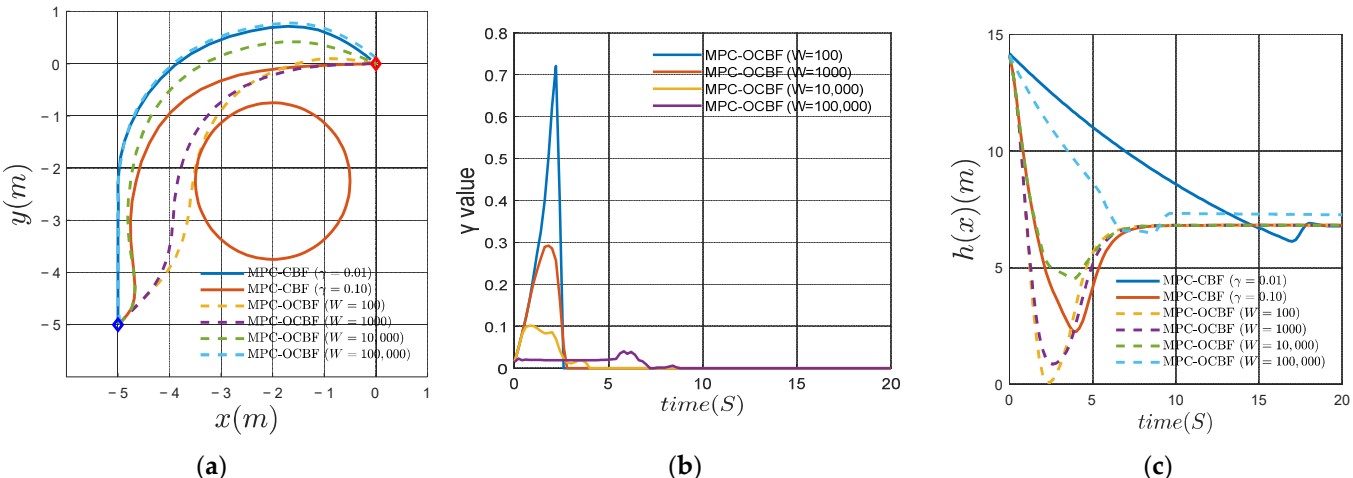

(a)                           (b)                           (c)

**Figure 1.** Compare the conservative obstacle avoidance behaviors generated by MPC-CBF and MPC-OCBF: (**a**) the trajectory of MPC-CBF and MPC-OCBF; (**b**) the optimized $\gamma$ value of MPC-OCBF with different weight matrices; (**c**) the CBF h(x) values of MPC-CBF and MPC-OCBF.

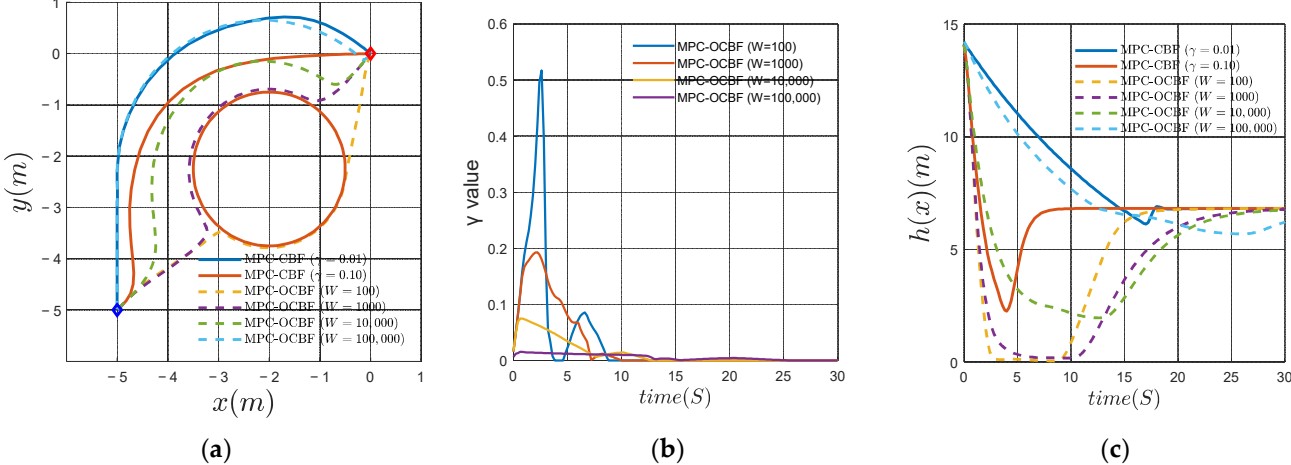

(a)                           (b)                           (c)

**Figure 2.** Compare the conservative obstacle avoidance behaviors generated by MPC-CBF ($N = 8$) and MPC-OCBF ($N = 2$): (**a**) the trajectory of MPC-CBF and MPC-OCBF; (**b**) the optimized $\gamma$ value of MPC-OCBF with different weight matrices; (**c**) the CBF h(x) values of MPC-CBF and MPC-OCBF.

As the relative important weight of $\gamma_t$ is increased in the cost represented by $W_\gamma$, the trajectory of MPC-OCBF becomes increasingly conservative, and at the same time, the optimized $\gamma$ values become smaller and smaller, as illustrated in Figure 1a,b. With different weights, various behaviors can be generated. As shown in Figure 1b, the $\gamma$ value adaptively becomes bigger to relax the safety restriction level when the conflict between different constraints exists. When the conflict vanishes, the $\gamma$ value becomes as small as possible.

In Figure 1c, the actual values of the CBF h(x) are given. As the $\gamma$ value is fixed In the MPC-CBF, its CBF curves are smoother than in our MPC-OCBF. Compared with MPC-CBF, our MPC-OCBF can change the $\gamma$ value at different time steps and thus its CBF curves can vary more flexibly and quickly to perform tasks.

Furthermore, the calculation time of MPC-CBF and MPC-OCBF is compared to analyze the increased complexity by adding another optimizable variable $\gamma$. Experiments were run on a notebook PC with 1.8 GHz CPU and 36 G RAM. The mean and standard deviation of the MPC-CBF optimization time of each time step were 21.6 ms and 6.7 ms, while the mean and standard deviation of the MPC-OCBF calculation time were 39.4 ms and 14.5 ms, respectively. Complexity was added by the new optimizable variable $\gamma$. Nevertheless, as our MPC-OCBF was able to adjust the $\gamma$ value adaptively, the time horizon in the MPC-OCBF could be reduced to accelerate the optimization. We further ran the MPC-OCBF with $N = 2$ without changing other parameters, and the results are illustrated in Figure 2.

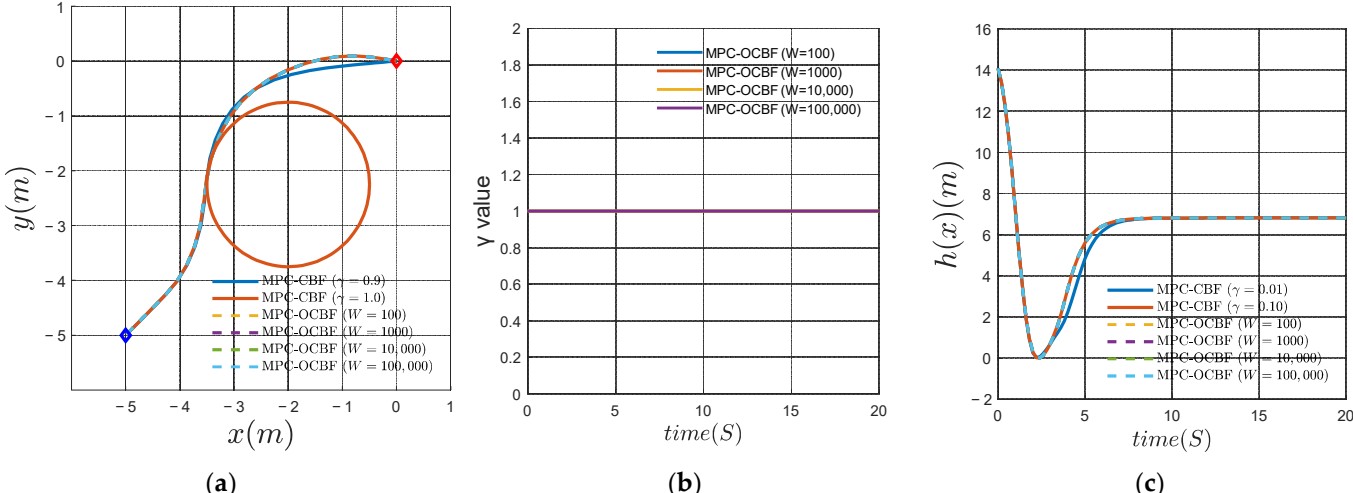

**Figure 3.** Compare the aggressive obstacle avoidance behaviors generated by MPC-CBF and MPC-OCBF: (**a**) the trajectory of MPC-CBF and MPC-OCBF; (**b**) the optimized $\gamma$ value of MPC-OCBF with different weight matrices; (**c**) the CBF h(x) values of MPC-CBF and MPC-OCBF.

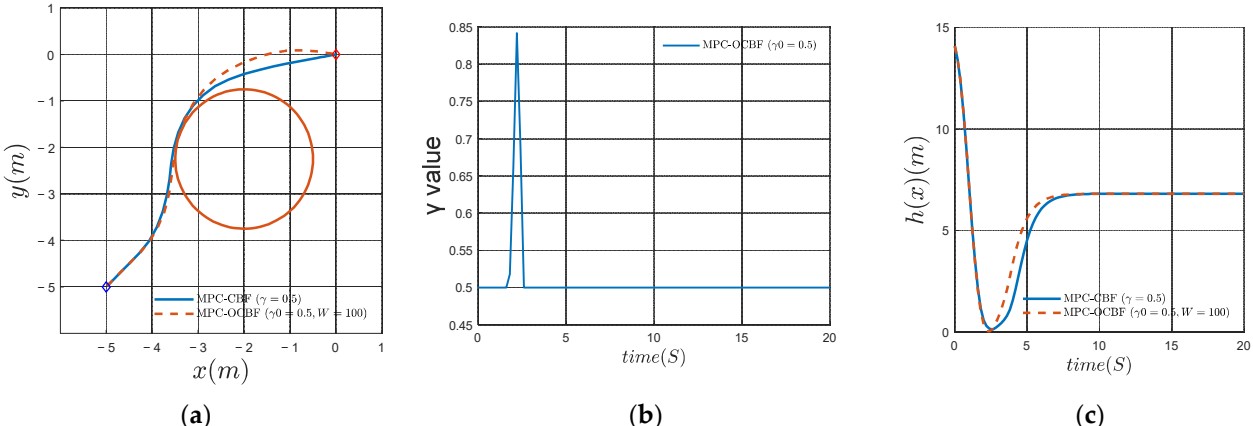

**Figure 4.** Compare the user-defined obstacle avoidance behaviors generated by MPC-CBF and MPC-OCBF: (**a**) the trajectory of MPC-CBF and MPC-OCBF; (**b**) the optimized $\gamma$ value of MPC-OCBF; (**c**) the CBF h(x) values of MPC-CBF and MPC-OCBF.

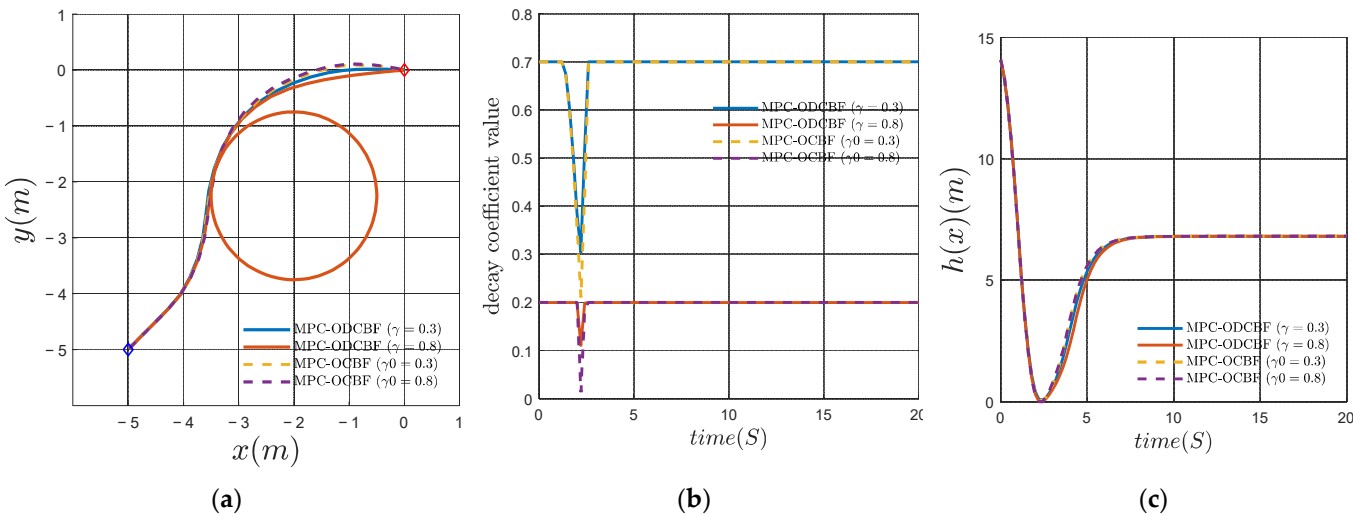

**Figure 5.** Compare the user-defined obstacle avoidance behaviors generated by MPC-ODCBF and MPC-OCBF: (**a**) the trajectory of MPC-ODCBF and MPC-OCBF; (**b**) the optimized decay coefficient of MPC-ODCBF and MPC-OCBF; (**c**) the CBF h(x) values of MPC-ODCBF and MPC-OCBF.

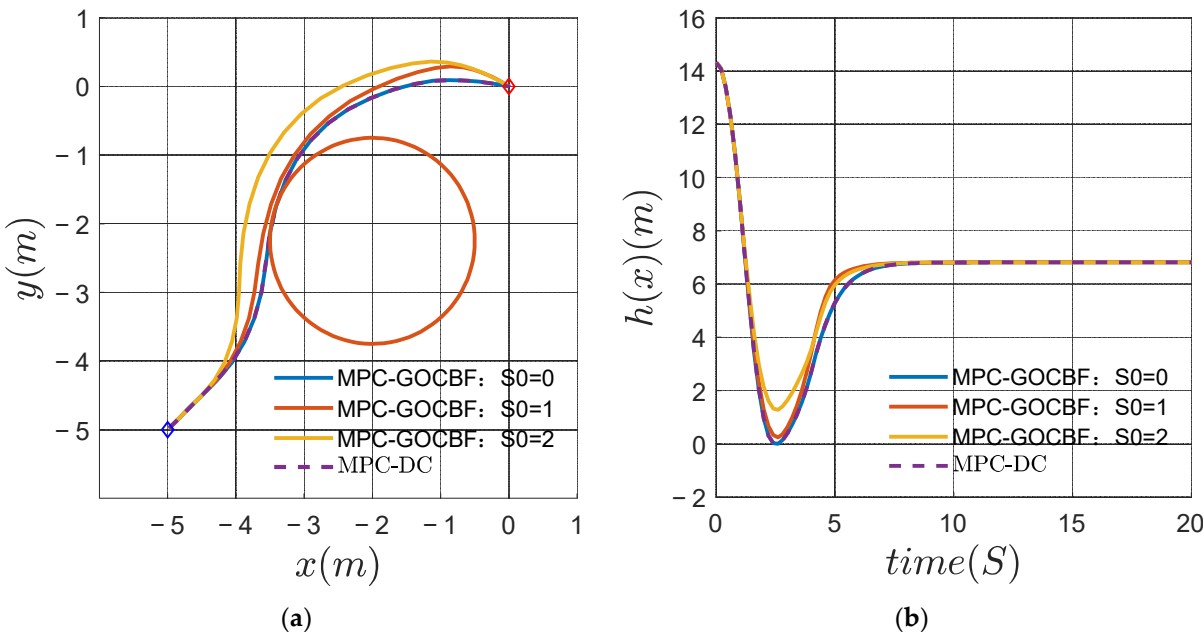

**Figure 6.** Compare the obstacle avoidance behaviors generated by MPC-DC and MPC-GOCBF: (**a**) the trajectory of MPC-GOCBF with different $S_0$ and MPC-DC; (**b**) the CBF h(x) values of MPC-GOCBF and MPC-DC.

Our MPC-OCBF can still produce various behaviors with the reduced MPC time horizon $N = 2$. However, for MPC-CBF, it has no feasible solution when $N = 2$. Nevertheless, it should be noted that our MPC-OCBF optimizes the $\gamma_t$ through a weighted multi-objection way. Consequently, the weights of different objections need to be fine-tuned to determine preferred behavior in real applications.

(2)  Prefer more aggressive behaviors

Equation (15b) is used in our MPC-OCBF, and the results are compared with MPC-CBF with fixed $\gamma_t$. For MPC-CBF, two experiments were run with $\gamma = 0.9$ and $\gamma = 1.0$, respectively. For MPC-OCBF, four experiments were run with the weight matrices $W_\gamma = 10^2 \cdot I_N$, $10^3 \cdot I_N$, $10^4 \cdot I_N$, and $10^5 \cdot I_N$, respectively. The results are shown in Figure 3.

The results of MPC-OCBF with different weight matrices are almost the same as illustrated in Figure 3a. These results are different to Figure 1a because the object of aggressive behaviors is consistent with other goal objects (the shortest path to reach the goal is going straight ahead) and consequently, there is no conflict between different constraints, thus the weight $W_\gamma = 10^2 \cdot I_N$ is already enough. As illustrated in Figure 3b, during optimization, the $\gamma$ value remains at 1 in our MPC-OCBF with different weight matrices. However, for conservative behaviors in Figure 1a, the safety object far from the obstacle is contrary to the object reaching the goal efficiently, thus a larger weight matrix is required to regulate different constraints.

(3) Prefer user-defined behaviors

Equation (15c) is used in the MPC-OCBF and the results are compared with MPC-CBF with the user-preferred $\gamma$ value. The user-preferred $\gamma$ value was set as $\gamma_0 = 0.5$. The MPC-OCBF is simulated with $W_\gamma = 10^2 \cdot I_N$. The results are illustrated in Figure 4.

Figure 4a shows that our MPC-OCBF can generate preferred behavior, as with the MPC-CBF, with the same fixed $\gamma$ value. It can be seen from Figure 4b that the optimized $\gamma$ value in MPC-OCBF remains as the user-defined value except in the stage near the obstacle. When the robot moves near the obstacle, the $\gamma$ value is increased to reduce the safety restriction level to consider other performance objects. When the obstacle avoidance task is finished, the $\gamma$ value is decreased back to the user-defined value, as illustrated in Figure 4b. Our MPC-OCBF framework can better coordinate conflicting objects by temporarily adapting the $\gamma$ value.

(4) Compare MPC-OCBF and MPC-ODCBF with user-defined behaviors

Our MPC-OCBF is further compared with the optimal-decay form of MPC-ODCBF under user-defined behaviors as illustrated in Figure 5, where two different preferred values $\gamma = [0.3, 0.8]$ were set. In MPC-ODCBF, the cost that tries to minimize $\sum p_w(\omega_k - 1)^2$ is added to make the $\gamma$ value close to user-defined values, where $p_w$ is the relative weight. For a fair comparison, $W_\gamma = 10^2 \cdot I_N$ and $p_w = 10^2$ were set, respectively, in MPC-OCBF and MPC-ODCBF.

The results of MPC-ODCBF and MPC-OCBF are almost the same in Figure 5. Moreover, the actual decay coefficient is plotted in Figure 5b: in MPC-ODCBF, it is $\omega_k(1 - \gamma)$; in the MPC-OCBF, it is $1 - \gamma$. For MPC-ODCBF, if the variation in $\omega_k$ is $\Delta\omega_k$, then the variation in the decay efficient is $\Delta\omega_k(1 - \gamma)$. For our MPC-OCBF, if the variation in $\gamma$ is $\Delta\gamma$, then the variation in the decay efficient is $\Delta\gamma$. Consequently, if the variation level is the same, i.e., $\Delta\omega_k = \Delta\gamma$, then $\Delta\omega_k(1 - \gamma) \leq \Delta\gamma$ because of the inequality relation $(1 - \gamma) < 1$. This is confirmed in Figure 5b as the results of our MPC-OCBF all have a greater change in amplitude than the MPC-ODCBF results in both examples. Compared with MPC-ODCBF, the optimization in our MPC-OCBF is clearer and more direct to reflect the flexibility to compromise conflicting objects.

(5) Compare MPC-DC with MPC-GOCBF

Finally, we compare our MPC-GOCBF with MPC-DC. Here, three different safety margin values $S_0 = [0,1,2]$ were set with the same $W_S = 10^2 \cdot I_N$. The results are illustrated in Figure 6.

The results of MPC-GOCBF with $S_0 = 0$ and MPC-DC are exactly the same, as illustrated in Figure 6a. The trajectory of $S_0 = 2$ is more conservative with a higher safety margin compared with the trajectory of $S_0 = 1$. The actual $h(x_t)$ values of different methods are presented in Figure 6b. It can be seen that the actual values of $h(x_t)$ are not always greater than $S_0$, which can be explained by the fact that our MPC-GOCBF can trade off different conflicting objects adaptively to enhance feasibility.

*5.2. Feasibility Region Test*

(1) Different initial states for the discrete-time simulation

To further test the feasibility performance of our framework, the discrete-time linear triple-integrator system in reference [4] was simulated with different initial conditions to test its feasibility level. Here, $x_{k+1} = Ax_k + Bu_k$, $x = [x, v, a]^T \in \mathbb{R}^3$, and $u = [j]^T$ represent position ($x$), velocity ($v$), acceleration ($a$), and jerk ($j$), respectively. The corresponding parameters used are the same as reference [4]. The CBF is defined as $h(x) = x^2 + v^2 + a^2 - 1$. Continuously changing the initial conditions facilitates testing of whether a feasible solution exists, where only the first time-step is tested. As with reference [4], we iterate over initial states in the closed space as $\mathbb{X} = \{-2 \leq x \leq 0, 0 \leq v \leq 2, 0 \leq a \leq 2\}$. The feasibility of different initial conditions of MPC-GCBF and our MPC-OCBF are illustrated in Figure 7, where the color dots represent feasibility.

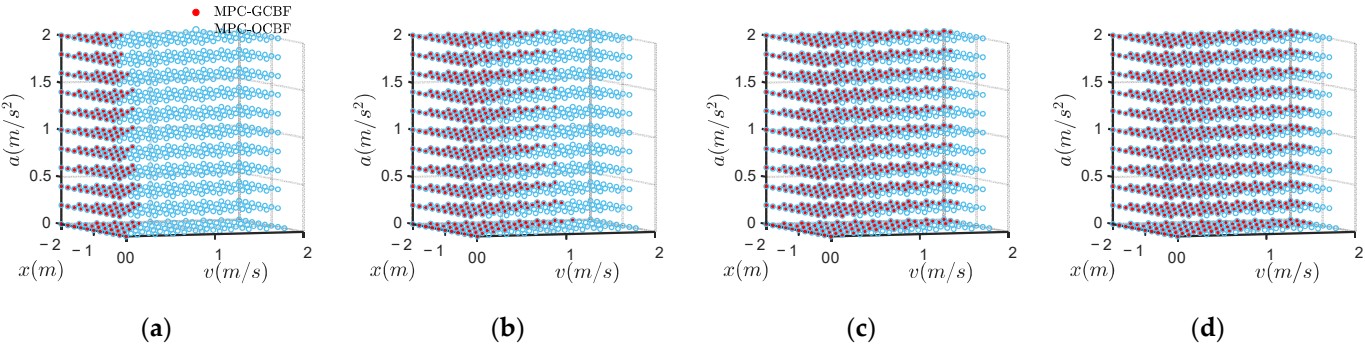

**(a)**        **(b)**        **(c)**        **(d)**

**Figure 7.** Feasibility comparison between MPC-OCBF and MPC-GCBF in the discrete-time formation: **(a)** $\gamma = 0.05$; **(b)** $\gamma = 0.10$; **(c)** $\gamma = 0.15$; **(d)** $\gamma = 0.20$.

This shows that the feasibility region of MPC-OCBF is independent of $\gamma$ and the feasibility region of MPC-GCBF is always a subset of MPC-OCBF, which confirms Remark 2, as detailed previously.

(2)     Different initial states for continuous simulation

The feasibility performance of our MPC-GOCBF was further tested using an adaptive cruise control (ACC) example, which is commonly used to validate safety-critical control strategies [5,13]. The dynamic of the system is given as:

$$\dot{x} = \begin{bmatrix} x_2 \\ -\frac{1}{m} F_r(x) \\ v_l - x_2 \end{bmatrix} + \begin{bmatrix} 0 \\ \frac{1}{m} \\ 0 \end{bmatrix} \tag{24}$$

where $(x_1, x_2)$ are the position and velocity ($x_2 = \dot{x}_1$) of the ego vehicle, $m$ is the mass of the vehicle, and $x_3$ is the distance between the ego and the lead vehicle traveling at a velocity of $v_l$. The aerodynamic drag is represented by $F_r = f_0 + f_1 x_2 + f_2 x_2^2$, where $f_0$, $f_1$ and $f_2$ are empirical constants. The CBF $h = x_3 - 1.8 x_2$ and the CLF $V = (x_2 - v_d)^2$, here $v_d$, represents the desired velocity of the ego vehicle. The input control constraint is added as $c_d mg \leq u \leq c_a mg$. As in reference [5], the MPC horizon $N = 1$ is used to enable fair comparison and it becomes the CBF-CLF-QP formulation. Other than extending the simulation time from 12 s to 30 s, the same parameters as reference [5] were used. Specifically, the ego vehicle is initialized at origin $x_1(0) = 0$, the initial distance between the ego car and the lead vehicle is set as $x_3(0) = 100$, and feasibility was tested with varying initial speeds $x_2(0)$ from 26 m/s to 32 m/s. Here, we compare the results of traditional CBF-CLF-QP using fixed $\gamma = 0.5$, optimal-decay CBF-CLF-QP in reference [5] with preferred value $\gamma = 0.5$ and our GOCBF-CLF-QP with the different user-preferred safety margin $S_0 = 25$ and $S_0 = 40$, respectively. The results are illustrated in Figure 8.

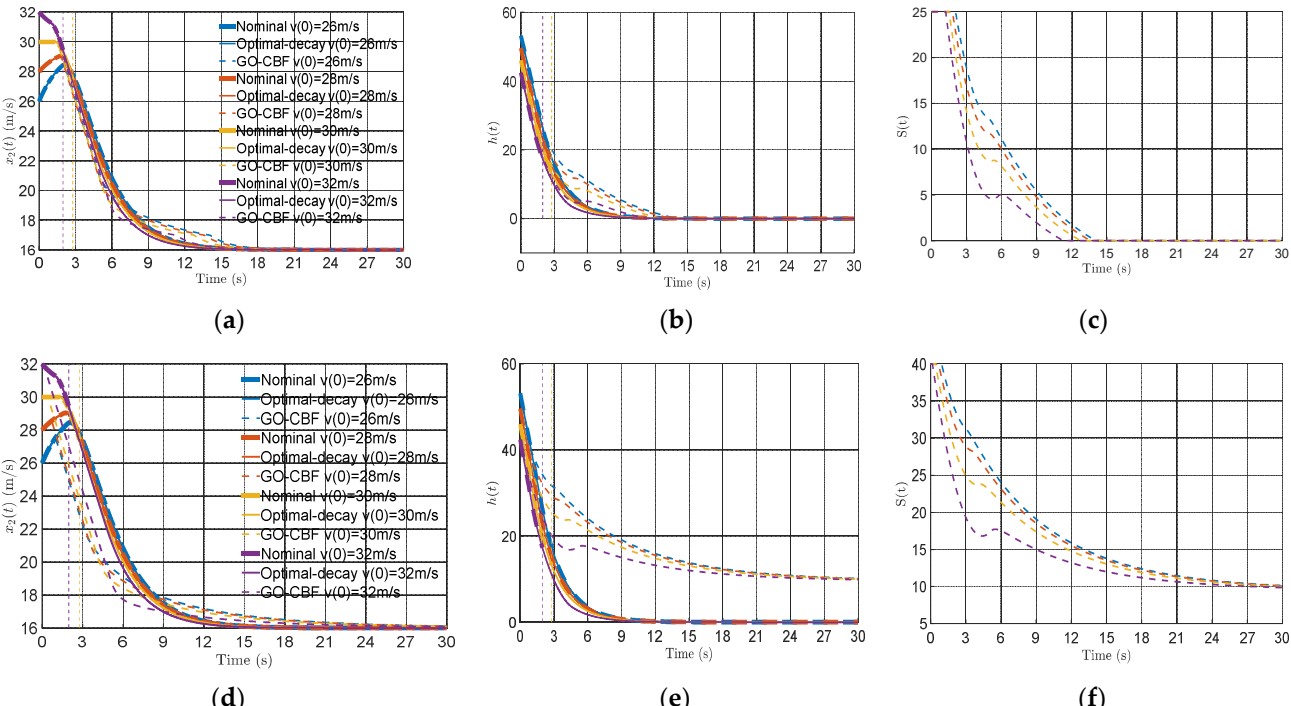

**Figure 8.** Simulation results of adaptive cruise control using fixed/optimal decay and our general and optimizable form of CLF-CBF-QP with different initial velocities. Different colors represent different initial conditions. For our general and optimizable form of CLF-CBF-QP, (**a**–**c**) are the results of $S_0 = 25$ and (**d**–**f**) are the results of $S_0 = 40$, while all of them use the same $W_S = 10^4$: (**a**) ego vehicle's speed; (**b**) values of control barrier function; (**c**) actual safety margin; (**d**) ego vehicle's speed; (**e**) the CBF h(x) values; (**f**) actual safety margin.

As illustrated in reference [5], the original CLF-CBF-QP with initial speed at 30 m/s and 32 m/s becomes infeasible during the simulations. This infeasibility derives from the conflict between the input constraint and original CBF constraint, while the optimal-decay CBF-CLF-QP [5] can enhance feasibility by using $\omega_k$ to adaptively change the decay coefficient. Our GOCBF not only enhances feasibility as the optimal-decay does, but goes further by generating different behaviors with different safety margins. Increasing $S_0$ from 25 to 40, the final CBF value can increase from above 0 to above 9.7, as illustrated in Figure 8b and Figure 8e, respectively. From the control view, it can be seen that the ego vehicle's speed decreases more quickly in $S_0 = 40$ than $S_0 = 25$ to satisfy the higher safety restriction level, with the cost that the CLF constraint is violated, as illustrated in Figure 8a and Figure 8d, respectively. It can thus be concluded that our MPC-GOCBF provides a more flexible and intuitive mechanism to compromise different objects.

### 5.3. Collision Avoidance among Multi-Agents

Finally, our MPC-GOCBF was tested in a more complex application of collision avoidance among multi-agents. Here, an ego agent is required to reach a fixed goal (a rectangle region) from the start while avoiding other dynamic agents, which are randomly initialized and walk at random, as illustrated in Figure 9.

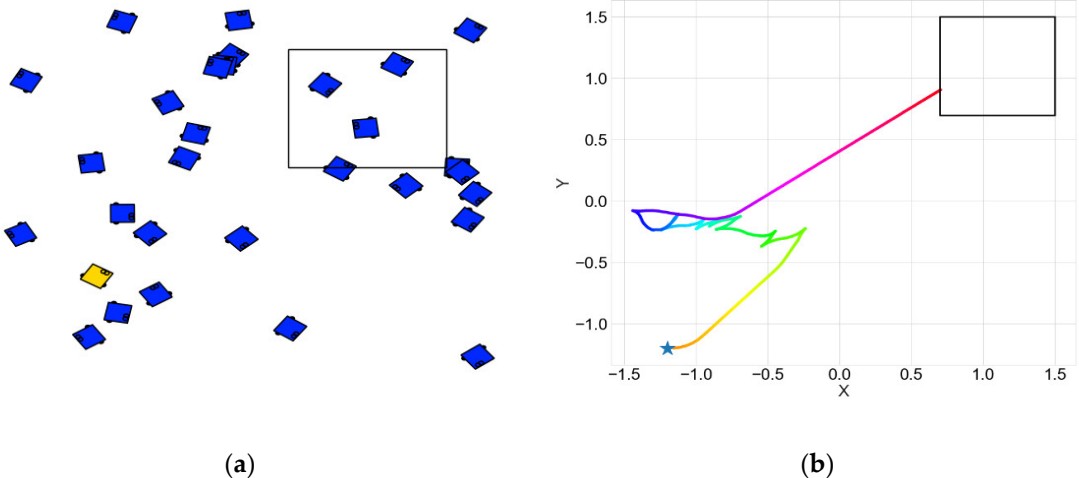

(**a**)          (**b**)

**Figure 9.** Collision Avoidance Among Multi-Agents. The ego agent is represented as yellow, while other agents used as dynamic obstacles are represented as blue: (**a**) a snapshot of the simulation; (**b**) a trajectory of the ego agent.

The state of the ego agent is represented as a three-tuple $(x(t), y(t), \theta(t))$, which are the 2D positions and angle, respectively. The linear and angle velocity $(v(t), \omega(t))$ are directly used as the controller input. Consequently, the dynamic of the ego agent can be represented as:

$$\begin{bmatrix} x(t+1) \\ y(t+1) \\ \theta(t+1) \end{bmatrix} = \begin{bmatrix} x(t) \\ y(t) \\ \theta(t) \end{bmatrix} + \begin{bmatrix} \cos\theta(t)v(t) \\ \sin\theta(t)v(t) \\ \omega(t) \end{bmatrix} \Delta t \tag{25}$$

The initial state of the ego agent is set as $(x(0), y(0), \theta(0)) = (-1.2, -1.2, 0)$ and the initial state of other dynamic agents is set randomly. Moreover, the control input of other dynamic agents is fixed as $(v(t), \omega(t)) = (0.15, 0.1)$ during simulation.

To compare the performance of our MPC-GOCBF with MPC-ODCBF, the following performance metrics were used:

**Success Rate**: the ratio of the number of the ego agents reaching their goal without any collisions with other agents.

**Average Distance**: the average traveled trajectory length of the ego agent over all the successful trajectories.

**Average Time**: the ego agent's average travel time over the successful trajectories.

The performance was evaluated over different numbers of agents, and for each method, every test case was evaluated for 100 repeats. The results are given in Table 1.

**Table 1.** Performance Metrics (As Mean/Std) Evaluated for MPC-ODCBF and our MPC-GOCBF with Different Numbers of Agents.

| Metric | Method | 20 | 30 | 40 | 50 |
|--------|--------|-----|-----|-----|-----|
| Success Rate | MPC − ODCBF $\gamma = 0.1$ | 60% | 58% | 54% | 47% |
| | MPC − ODCBF $\gamma = 0.9$ | 96% | 89% | 78% | 82% |
| | MPC-GOCBF $S_0 = 0.01$ | 98% | 97% | 94% | 85% |
| | MPC-GOCBF $S_0 = 0.05$ | 96% | 92% | 91% | 89% |
| Average Time | MPC − ODCBF $\gamma = 0.1$ | 87.85/33.36 | 96.43/37.62 | 109.93/28.44 | 114.51/31.54 |
| | MPC − ODCBF $\gamma = 0.9$ | 104.24/43.38 | 115.06/37.61 | 138.38/52.39 | 138.61/40.42 |
| | MPC-GOCBF $S_0 = 0.01$ | 102.31/41.34 | 116.55/40.63 | 120.87/36.10 | 139.36/37.74 |
| | MPC-GOCBF $S_0 = 0.05$ | 93.88/41.31 | 113.97/39.38 | 123.69/36.66 | 133.39/1.04 |
| Average Distance | MPC − ODCBF $\gamma = 0.1$ | 3.62/1.19 | 3.76/1.26 | 4.16/1.00 | 4.09/0.90 |
| | MPC − ODCBF $\gamma = 0.9$ | 4.13/1.64 | 4.45/1.40 | 5.12/1.97 | 4.83/1.35 |
| | MPC-GOCBF $S_0 = 0.01$ | 3.99/1.48 | 4.34/1.32 | 4.46/1.22 | 4.83/1.25 |
| | MPC-GOCBF $S_0 = 0.05$ | 3.75/1.39 | 4.25/1.35 | 4.45/1.34 | 4.55/1.04 |

To test the statistical difference between the proposed MPC-GOCBF algorithm and the compared MPC-ODCBF, a one-way analysis of variance (ANOVA) test [44] was applied based on Performance Metrics Average Time, and Average Distance, respectively. The ANOVA test results are shown in Tables 2 and 3.

**Table 2.** ANOVA Test Results Based on Performance Metrics Average Time.

| $N = 20$ | | | | | | |
|---|---|---|---|---|---|---|
| **Source of Variance** | **SS** | **DF** | **MS** | **F (DFn, DFd)** | ***p*-value** | **F crit** |
| **Between Group** | 13,368.813 | 3 | 4456.271 | 2.694 | 0.046 | 2.631 |
| **Within Group** | 572,304.456 | 346 | 1654.059 | | | |
| **Total** | 585,673.267 | 349 | | | | |
| $N = 30$ | | | | | | |
| **Source of Variance** | **SS** | **DF** | **MS** | **F (DFn, DFd)** | ***p*-value** | **F crit** |
| **Between Group** | 17,250.872 | 3 | 5750.291 | 3.782 | 0.011 | 2.631 |
| **Within Group** | 504,757.887 | 332 | 1520.355 | | | |
| **Total** | 522,008.759 | 335 | | | | |
| $N = 40$ | | | | | | |
| **Source of Variance** | **SS** | **DF** | **MS** | **F (DFn, DFd)** | ***p*-value** | **F crit** |
| **Between Group** | 27,758.935 | 3 | 9252.978 | 5.834 | 0.000688 | 2.633 |
| **Within Group** | 496,406.018 | 313 | 1585.962 | | | |
| **Total** | 524,164.953 | 316 | | | | |
| $N = 50$ | | | | | | |
| **Source of Variance** | **SS** | **DF** | **MS** | **F (DFn, DFd)** | ***p*-value** | **F crit** |
| **Between Group** | 22,012.770 | 3 | 7337.590 | 5.746 | 0.000781 | 2.635 |
| **Within Group** | 381,796.187 | 299 | 1276.910 | | | |
| **Total** | 403,808.957 | 302 | | | | |

**Table 3.** ANOVA Test Results Based on the Performance Metrics Average Distance.

| $N = 20$ | | | | | | |
|---|---|---|---|---|---|---|
| **Source of Variance** | **SS** | **DF** | **MS** | **F (DFn, DFd)** | ***p*-value** | **F crit** |
| **Between Group** | 12.923 | 3 | 4.308 | 2.034 | 0.109 | 2.631 |
| **Within Group** | 732.641 | 346 | 2.117 | | | |
| **Total** | 745.564 | 349 | | | | |
| $N = 30$ | | | | | | |
| **Source of Variance** | **SS** | **DF** | **MS** | **F (DFn, DFd)** | ***p*-value** | **F crit** |
| **Between Group** | 18.325 | 3 | 6.108 | 3.417 | 0.0177 | 2.632 |
| **Within Group** | 593.547 | 332 | 1.788 | | | |
| **Total** | 611.872 | 335 | | | | |
| $N = 40$ | | | | | | |
| **Source of Variance** | **SS** | **DF** | **MS** | **F (DFn, DFd)** | ***p*-value** | **F crit** |
| **Between Group** | 35.344 | 3 | 11.781 | 5.640 | 0.000893 | 2.633 |
| **Within Group** | 653.783 | 313 | 2.089 | | | |
| **Total** | 689.126 | 316 | | | | |
| $N = 50$ | | | | | | |
| **Source of Variance** | **SS** | **DF** | **MS** | **F (DFn, DFd)** | ***p*-value** | **F crit** |
| **Between Group** | 20.837 | 3 | 6.946 | 5.034 | 0.00203 | 2.635 |
| **Within Group** | 412.517 | 299 | 1.380 | | | |
| **Total** | 433.355 | 302 | | | | |

From Table 2, it can be seen that there is a statistically significant difference between the means of the four groups of Average Time (all *p*-values in Table 2 are less than 0.05.),

which confirms the time-complexity analysis of our MPC-GOCBF and MPC-CBF. In Table 3, when $N = 20$, the $p$-value is 0.109, which is greater than 0.05. Nevertheless, all other $p$-values in Table 3 are less than 0.05 except $N = 20$. This is coincident with our intuition. When the number of dynamic obstacles is small, the Average Distance of different methods is almost the same because the collision avoidance behaviors are few and almost the same. However, when the number of dynamic obstacles becomes larger, the Average Distance of different methods varies significantly, because the collision avoidance behaviors also vary significantly and there are more of them.

Table 1 shows that as the number of other dynamic agents increases, the success rate decreases because the problem becomes more difficult. Moreover, if the safety restriction level is relaxed (increasing $\gamma$ in MPC-ODCBF or decreasing $S_0$ in MPC-GOCBF), the success rate will increase for both methods, which indicates that, in this example, the safety restriction level and feasibility are in conflict. Nevertheless, it can be seen that our MPC-GOCBF has a more intuitive and clear meaning with the safety restriction level set by $S_0$, which is more convenient for real applications. Moreover, to analyse the sensitivity [45] of the parameter $N$ (the number of dynamic obstacles) on the final performance metric Success Rate, the Change Rates of different methods on Success Rate ($N = 20$) are given in Figure 10. As the number of obstacles increases, compared with MPC-ODCBF, it can be seen that the performance of our MPC-GOCBF remains more stable with the same hyper-parameter $S_0$ as the number of dynamic obstacles increases.

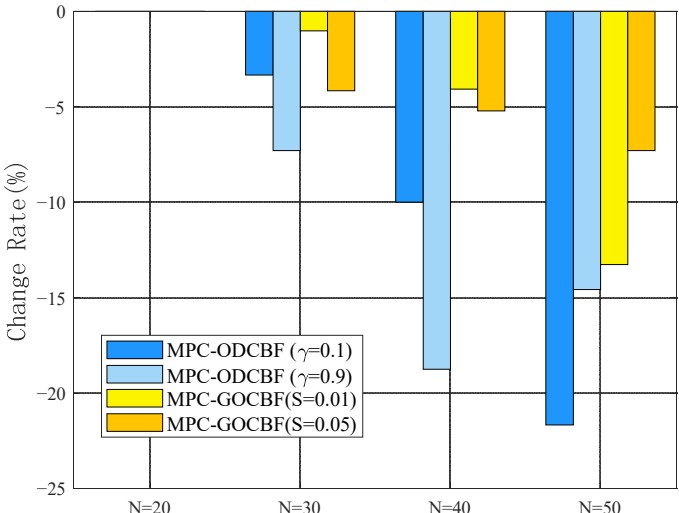

**Figure 10.** The change rate of our MPC-GOCBF with MPC-ODCBF as the number of dynamic obstacles $N$ increases from 20 to 50, with $N = 20$ as the base.

## 6. Conclusions

In this paper, we propose a general and optimizable CBF formulation that can enhance feasibility by flexibly adapting the safety restriction level according to actual situations. In this formulation, the safety restriction level defined by our OCBF or GOCBF can be adaptively and flexibly adjusted to resolve conflicts between feasibility and safety. Furthermore, the safety margin defined in our GOCBF has intuitive physical meanings, meaning that the method proposed in this paper provides a more flexible and intuitive mechanism to compromise different objects compared with existing formulations in terms of ensuring safety while not undermining basic feasibility.

Future work should consider real system applications that may have modeling errors, system disturbance or noise. Moreover, the proposed formulation can be combined with data-driven machine-learning methods to adapt to more complex situations.

**Author Contributions:** Conceptualization and methodology, S.L.; software, validation, formal analysis, investigation, resources, and data curation, S.L., Z.Y. (Zhimin Yuan), Y.C., F.L., Z.Y. (Zhichao Yang)

and Q.Y.; writing—original draft preparation, S.L.; writing—review and editing, S.L., W.F. and Y.F.; visualization and supervision, Q.Y., W.F. and Y.F. All authors have read and agreed to the published version of the manuscript.

**Funding:** This research was funded by National Defense Science and Technology Foundation Enhancement Plan with grant No. 2019-JCJQ-JJ-042, the China Postdoctoral Science Foundation with grant No. 2014M562555 and National Natural Science Foundation of China with grant No. 61202338.

**Acknowledgments:** We thank all anonymous reviewers for their helpful comments and constructive feedbacks on an earlier draft of this article. Great thanks to Jun Zeng for providing the source code and experiment results of MPC-CBF in https://github.com/HybridRobotics/MPC-CBF, accessed on 1 October 2022.

**Conflicts of Interest:** The authors declare no conflict of interest. The funders had no role in the design of the study; in the collection, analysis, or interpretation of data; in the writing of the manuscript; or in the decision to publish the results.

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
