# Peer review of "Optimizable Control Barrier Functions to Improve Feasibility and Add Behavior Diversity while Ensuring Safety"

_electronics, doi:10.3390/electronics11223657_

Round 1

Reviewer 1 Report

The title should be updated.
Enhance the introduction to show the motivation of this work.
It is better to present the suggested method in an algorithmic form.
The Abstract is suggested to be improved where the contribution and findings of the work should be highlighted.
Enhance the English of the work. There are too many problems with paper typesetting.
How to initialize the agents in the proposed Algorithm?
It is necessary to discuss the complexity of the proposed solution.
Statistical analysis should be carried out to demonstrate that the experimental results are significant. Such as the ANOVA test.
You can find Statistical analysis for this reference :

Takieldeen, Ali E., El-Sayed M. El-kenawy, Mohammed Hadwan, and Rokaia M. Zaki. "Dipper throated optimization algorithm for unconstrained function and feature selection." Comput., Mater. Continua 72, no. 1 (2022): 1465-1481.

Reviewer 2 Report

The article proposes an Optimizable CBF (OCBF). Authors extend their version to a General OCBF (GOCBF) formulation to eliminate the accumulation effects caused by the actual values of CBF constraints in previous time steps.

They suppose the CBF constraints are regarded as optimizable parameters within the MPC framework rather than predefined and fixed values.

According to their hypothesis, the safety constraint level can be defined intuitively and adjusted flexibly using the proposed formula. Experimental results demonstrate the algorithm's capabilities to provide a more flexible and intuitive mechanism for compromising various objects, thereby improving feasibility and adding behavioural diversity to the MPC-CBF framework.

The research concepts of the proposed mechanisms are validated with experimental results and the application of mathematical formulations.

The article has been written following good scientific practice and according to the publishers' policy. The results are presented clearly and convincingly.

Author Response

We are very grateful to the reviewer for this accurate summary and the kind recognition of our key contributions.

We have enhance the English of the work as best as we can.

Thank you greatly for taking time to review our paper.  

Round 2

Reviewer 1 Report

It is necessary to discuss the complexity of the proposed solution.

If you see "no need to do the ANOVA test," SO add a T-Test to show that experimental results are significantly different between the two groups.
Some additional experiments are required, as Sensitivity analysis

you can check all from this reference 
El-kenawy, E.-S.M.; Albalawi, F.; Ward, S.A.; Ghoneim, S.S.M.; Eid, M.M.; Abdelhamid, A.A.; Bailek, N.; Ibrahim, A. Feature Selection and Classification of Transformer Faults Based on Novel Meta-Heuristic Algorithm. Mathematics 2022, 10, 3144. https://doi.org/10.3390/math10173144

Round 3

Reviewer 1 Report

Well Done, thanks for your efforts

All revision notes have been done